# Tick-Borne Encephalitis Specific Lymphocyte Response after Allogeneic Hematopoietic Stem Cell Transplantation Predicts Humoral Immunity after Vaccination

**DOI:** 10.3390/vaccines9080908

**Published:** 2021-08-15

**Authors:** Nicole Harrison, Katharina Grabmeier-Pfistershammer, Alexandra Graf, Doris Trapin, Peter Tauber, Judith H. Aberle, Karin Stiasny, Ralf Schmidt, Hildegard Greinix, Werner Rabitsch, Michael Ramharter, Heinz Burgmann, Winfried F. Pickl, Christina Bahrs

**Affiliations:** 1Division of Infectious Diseases and Tropical Medicine, Department of Medicine I, Medical University of Vienna, 1090 Vienna, Austria; nicole.harrison@meduniwien.ac.at (N.H.); heinz.burgmann@meduniwien.ac.at (H.B.); 2Division of Cellular Immunology and Immunohematology, Institute of Immunology, Center for Pathophysiology, Infectiology and Immunology, Medical University of Vienna, 1090 Vienna, Austria; katharina.grabmeier-pfistershammer@meduniwien.ac.at (K.G.-P.); doris.trapin@meduniwien.ac.at (D.T.); peter.tauber@meduniwien.ac.at (P.T.); winfried.pickl@meduniwien.ac.at (W.F.P.); 3Section of Medical Statistics, Center for Medical Statistics, Informatics and Intelligent Systems, Medical University of Vienna, 1090 Vienna, Austria; alexandra.graf@meduniwien.ac.at; 4Center for Virology, Medical University of Vienna, 1090 Vienna, Austria; judith.aberle@meduniwien.ac.at (J.H.A.); karin.stiasny@meduniwien.ac.at (K.S.); 5Division of Virology, Department of Laboratory Medicine, Medical University of Vienna, 1090 Vienna, Austria; ralf.schmidt@meduniwien.ac.at; 6Division of Hematology, Department of Internal Medicine, Medical University of Graz, 8036 Graz, Austria; hildegard.greinix@medunigraz.at; 7Bone Marrow Transplantation Unit, Department of Medicine I, Medical University of Vienna, 1090 Vienna, Austria; werner.rabitsch@meduniwien.ac.at; 8Department of Tropical Medicine, Bernhard Nocht Institute for Tropical Medicine & I. Department of Medicine, University Medical Center Hamburg-Eppendorf, 20359 Hamburg, Germany; ramharter@bnitm.de; 9Institute of Infectious Diseases and Infection Control, Jena University Hospital/Friedrich-Schiller University, 07747 Jena, Germany

**Keywords:** allogeneic stem cell transplant, tick-borne encephalitis vaccination, lymphocyte proliferation, cytokine response, interleukin 13, sibling donors, humoral responders

## Abstract

The aim of this prospective study was to assess lymphocyte proliferative and cytokine response prior to and following tick-borne encephalitis (TBE) immunization among patients after allogeneic hematopoietic stem cell transplantation (HSCT). Seventeen adult patients 11–13 months after HSCT and eight unvaccinated healthy adults received up to three TBE vaccinations. Following in vitro stimulation with TBE-antigen, lymphocyte proliferation and cytokine secretion (IL-2, IL-10, IL-13, TNF-alpha, IFN-gamma, GM-CSF) were analyzed by thymidine incorporation assay and the Luminex system. Ten patients (59%) showed significant baseline TBE-specific lymphocyte proliferation (stimulation index (SI) > 3) prior to vaccination, but none of the unvaccinated controls (*p* = 0.002). All patients with a TBE-specific antibody response after two vaccinations (at least 2-fold increase of neutralization test titers) exhibited a strong TBE-specific lymphocyte proliferative response at baseline (SI > 10). Patients with sibling donors had a significantly stronger baseline TBE-specific lymphocyte proliferative and IL-13 cytokine response than patients with unrelated donors (*p* < 0.05). In conclusion, a relevant proportion of patients showed TBE-specific lymphocyte proliferative and cytokine responses prior to vaccination after HSCT, which predicted the humoral response to the vaccine. Patients with vaccinated sibling donors were more likely to elicit a cellular immune response than patients with unrelated donors of unknown vaccination status.

## 1. Introduction

Tick-borne encephalitis (TBE), caused by tick-borne encephalitis virus (TBEV), is the most common viral disease transmitted by infected ticks in Austria and many other areas in Central and Eastern Europe [1]. In recent years, TBE has emerged in new regions [1] and shown a steady increase of cases in countries like Germany and Slovenia according to the European Center of Disease Control and Prevention [2].

Patients with immunosuppressive conditions might suffer from an increased risk of severe disease. Considering the limited treatment options, prevention by active immunization is of crucial importance to avoid TBEV-related injury to the nervous system or even death [3,4]. Patients after allogeneic hematopoietic stem cell transplantation (HSCT) suffer from immunosuppression due to delayed immune reconstitution, sustained immunosuppressive medication, and underlying graft-versus-host disease and can therefore be considered as especially vulnerable to infectious diseases [5,6,7].

Our recent data showed a significantly reduced humoral response to TBE-vaccination in allogeneic HSCT recipients assessed by neutralization assay four weeks after the second vaccination with the adjuvanted inactivated whole virus vaccine FSME Immun^®^ compared to healthy controls [8]. Immune reconstitution takes several months to years after HSCT and an effective vaccine response might be hampered by an insufficient number of mature lymphocytes [5,9,10]. While the humoral response by measuring neutralizing antibodies is the FDA-approved primary endpoint of immunogenicity for flavivirus vaccines [11], the cellular response appears to be of particular interest in patients with an impaired immune system.

Our recent study on the unfolding humoral immunity to TBE-vaccination in allogeneic HSCT recipients has shown that the total number of CD4+ T-cells prior to re-vaccination was a significant predictor for neutralizing antibody production after two vaccinations [8]. In another clinical study by Aberle et al. that compared human CD4+ T-cell responses after TBE-vaccination and infection, cytokine patterns after vaccination differed from those after infection [12]. Specific responses during TBEV infection displayed a Th1 cytokine pattern with raised interferon-gamma (IFN-γ), interleukin 2 (IL-2) and tumor necrosis factor-alpha (TNF-α), whereas in vaccinated individuals CD4+ T-cells produced low amounts of IFN-γ [12]. The majority of CD4+ T-cells after vaccination expressed an IL-2 + TNF-α + IFN-γ-phenotype, which has been described for other protein-based vaccines as well [13]. So far, lymphocyte proliferative and cytokine responses after TBE-vaccination in HSCT recipients remained understudied. Moreover, it is currently unclear if a neutralizing antibody titer (NT) ≥ 10, as recommended by the WHO [14], may be sufficient for protection in this vulnerable group of patients. Accordingly, assessing the immune response not only on a humoral but also on a cellular level appeared to be of utmost importance to gain further insights into the vaccination response of HSCT patients.

The aim of this study was to assess the cellular immune response to TBE vaccination in allogeneic HSCT recipients one year after transplantation by analyzing lymphocyte proliferative and cytokine response at baseline, and after two and three TBE vaccinations.

## 2. Materials and Methods

### 2.1. Study Population and Design

In this prospective single-center open-label study, adult patients ≥ 18 years of age were enrolled 11–13 months after allogeneic HSCT at the Outpatient Clinic of the Bone Marrow Transplant Unit of the University Hospital of Vienna, Austria. Healthy controls ≥ 18 years of age without prior TBE vaccination were recruited at the Medical University of Vienna. Inclusion and exclusion criteria have been previously published by Harrison et al. [8].

Patients and healthy controls received up to three doses of FSME Immun^®^ intramuscularly—first at baseline at enrollment, which took place 11–13 months after HSCT, second after four weeks and third after 9–12 months. Each dose of FSME Immun^®^ contains 2.4 µg of formalin-inactivated TBE antigen (with the major constituent virus envelope protein E [15]) of the European TBEV subtype strain Neudörfl, human albumin as stabilizer and aluminum hydroxide as an adjuvant. Peripheral blood mononuclear cells (PBMCs) were sampled at baseline before the first vaccination, one week after the second and one week after the third vaccination.

### 2.2. Institutional Review Board Statement

Participation was voluntary and all participants gave written informed consent before participating in the study. The study was conducted in accordance with the Declaration of Helsinki. The study protocol was approved by the Ethics Committee of the Medical University of Vienna (No. 830/2011) and by the Austrian Competent Authorities represented by the Agency for Health and Food Safety (AGES PharmMed). This study was registered with clinicaltrials.gov (NCT01991067).

### 2.3. Study Endpoints

The study endpoints on cellular immunity reported in this manuscript are all secondary endpoints as the results of the primary endpoint, the humoral antibody response 4 weeks after the second vaccination, have already been published previously [8]. Secondary outcome measures of cellular immunity were lymphocyte proliferation upon stimulation with TBE antigen and quantitative levels of the cytokines IL-2, IFN-γ, IL-13, IL-10, TNF-α), and GM-CSF at baseline before vaccination, 7 days after the second and the third vaccination in patients versus healthy controls. Additionally, subgroup analyses of lymphoproliferative and cytokine responses were performed in HSCT patients comparing humoral responders (who had an NT ≥ 10 and at least a 2-fold increase of NTs after two vaccinations from baseline or an NT above the highest level of measurement) and non-responders, as well as patients with related and unrelated donors. Predictors for TBE-specific lymphocyte proliferative response at baseline were assessed in patients only.

### 2.4. Laboratory Analyses

#### 2.4.1. Collection and Storage of Peripheral Blood Mononuclear Cells

Peripheral blood mononuclear cells (PBMCs) were isolated from heparinized venous blood by Ficoll centrifugation at all three time points. The harvested PBMCs were added to a medium containing RPMI-1640 (Sigma-Aldrich, St. Louis, NA, USA) and 10% fetal calf serum (Gibco, Thermo Fisher Scientific, Waltham, Massachusetts, MA, USA) supplemented with penicillin (1%) and amphotericin (0.5%). The samples were cryopreserved and stored in liquid nitrogen.

#### 2.4.2. Lymphocyte Proliferation Detected by Thymidine Incorporation Assay

PBMCs (1 × 10^5^/well) were incubated in triplets at 37 °C and 5% CO_2_ in 96-well round bottom plates with the aluminum hydroxide-free and human albumin-free TBE antigen (FSME Ticovac strain Neudörfl, Pfizer, 0.6 µg/mL), the CEF MHC-I control peptide pool (Anaspec, 32 peptides, 2.5 µg/mL per peptide), a positive control peptide mix consisting of CEFT MHC-II peptide pool (JPT Peptides, 14 peptides, 0.25 µg/mL per peptide, each corresponding to a defined HLA class II restricted T-cell epitope from cytomegalovirus (CMV), Epstein–Barr virus (EBV), influenza virus or *Clostridium tetani*), tetanus toxoid (Statens Serum Institute, Copenhagen, Denmark, 0.125 Lf/mL) and tuberculin purified protein derivative PPD (Statens Serum Institute, Copenhagen, Denmark, 0.5 µg/mL); further, a mix consisting of phorbol myristate acetate (Sigma, 1.25 × 10^−8^ M) and ionomycine (Sigma, 0.1 µg/mL) or medium alone (RPMI, HyClone plus 2% Human Serum, Sigma). After 120 h cells were pulsed with methyl-[3H]thymidine (1 µCi/well) for 18 h and T-cell proliferation was quantified on a MicroBeta2 Microplate Counter (PerkinElmer, Waltham, MA, USA) as counts × 1000/min. Data was then standardized based on unstimulated controls of each patient.

#### 2.4.3. Flow Cytometric Analysis

The following lymphocyte populations were determined by immunofluorescence staining and flow cytometric analysis (FACS) from unstimulated cells of patients at baseline: lymphocytes, T-lymphocytes (CD3+), T helper cells (CD4+), naïve and memory T helper cells (CD4+CD45RA+ and RO+), T suppressor cells (CD8+), naïve and memory T suppressor cells (CD8+CD45RA+ and RO+), B-lymphocytes (CD19+) and B cell subsets (CD19+CD21low immature B cells, CD19+CD21highCD38+IgMhigh transitional B cells, CD19+CD10-CD27-CD21high naïve B cells, CD19+CD27+IgD+ and CD19+CD27+IgD- non class-switched and class-switched memory B cells, and plasmablasts). In addition, to assess immune reconstitution of patients over the study period, lymphocyte subpopulations (CD3+, CD4+, CD8+ and CD19+) were analyzed at 12 weeks after the first vaccination and at the time of the third vaccination.

#### 2.4.4. Cytokine Detection Assays

Supernatants of PBMC cultures incubated with the indicated stimuli (see above) were harvested after 96 h and subjected to multiplex cytokine analyses. Determination of secreted cytokines was performed using the Luminex system (Luminex 100IS, Biomedica, Vienna, Austria) consisting of a panel of 7 different bead regions from Luminex and antibody pairs from eBioscience or Biolegend (Appendix A). Four million Luminex microspheres had previously been coupled with 100 µg capture antibody (one antibody specificity per bead region). Collected supernatants were incubated with the capture antibody-coupled bead mixture in Multiscreen filterplates (MultiScreen^®^ HTS BV, Merck Millipore, Burlington, MA, USA) overnight at 4 °C in the dark on a lab dancer. After washing the microspheres with PBS, the biotinylated detection antibody mix was added at a final concentration of 2.5 µg/mL and the plates incubated at room temperature for two hours. After washing the microspheres with PBS, Streptavidin-Phycoerythrin Conjugate (eBioscience) was added at a final concentration of 2 µg/mL for 3 min. After washing with PBS, beads were resuspended in PBS and the mean fluorescent intensity (MFI) was measured for every bead region (analyte) at the Luminex 100IS device. Standards were bought from Peprotech Ltd and Biolegend. The standard curve comprised 16 data points and ranged from 10,000 to 0.61 pg/mL in a 2-fold dilution. Data was then standardized based on unstimulated controls of each patient.

#### 2.4.5. Statistical Analyses

Descriptive statistics (median, 1st and 3rd quartile) were calculated separately for patients and controls as well as overall. Cellular data (TBE proliferation, CEFT proliferation, PMA iono proliferation, and individual cytokines) of patients and controls were compared separately for time points using Wilcoxon-Rank-Sum-Tests. Within the group of patients, descriptive statistics were calculated separately for humoral responders and non-responders as well as for patients with related and unrelated donors. Cellular data for each time point were compared between groups using Wilcoxon-Rank-Sum-Tests. Univariable logistic regression models with Firth correction were calculated to evaluate the influence of age, body mass index (BMI), sex, donor, immunosuppressive medication, and total number of CD4+ T-cells on the baseline TBE proliferation values of patients only. Fisher Exact tests were calculated to investigate the relation between relevant amounts of cytokines and response. All analyses were performed using R, release 4.0.3.

## 3. Results

### 3.1. Characteristics of the Study Population

From July 2014 to January 2018, 19 patients and 15 healthy controls were enrolled into the study as described previously [8]. Of these, 17 patients and 8 healthy control subjects, from whom sufficient PBMCs could be cryopreserved to analyze cellular data, were included into the present follow-up study. Demographics of patients and control subjects are shown in Table 1. Control subjects had a median age of 29.5 (range 21; 60) compared to a median age of 31 years (range 22; 61) in the patient collective. Pre- and posttransplant characteristics of patients and TBE immunization data of patients and sibling donors prior to transplantation have been published previously [8]. In summary, all patients except one and all sibling donors had received a complete basic TBE vaccination schedule and at least one booster vaccine before HSCT (average time from last booster vaccine to HSCT/donation: 83.5 months for patients, 43.5 months for donors). In addition, immune reconstitution of patients was assessed by analyzing lymphocyte subpopulations at three different time points, which was in part published previously [8] and is added in the Appendix A.

### 3.2. Assessment of Lymphocyte Proliferation and Cytokine Responses after Antigen-Specific and Polyclonal Stimulation

#### 3.2.1. Comparison between Patients and Healthy Control Subjects

Proliferation of PBMCs obtained from study subjects at baseline are shown in Table 1. Healthy controls without prior TBE vaccination did not show any significant lymphocyte proliferation upon incubation with TBE antigen (stimulation index (SI) < 3), whereas they showed vigorous proliferation upon incubation with the polyclonal, pharmacological stimulus PMA plus ionomycin as well as with the antigen mix consisting of tetanus toxoid, PPD and CEF peptide mix (Figure 1 and Appendix A). In total, 10 out of 17 HSCT patients presented with baseline TBE proliferation (cut-off SI ≥ 3). Consistently, PBMCs of these patients were also reactive to the antigen mix containing tetanus toxoid, PPD, and peptides derived from CMV, EBV and influenza. Of the seven patients whose PBMCs did not show any baseline proliferation to TBE antigen, only the PBMCs of one patient presented with a modest proliferative response after TBE re-vaccination (subject #7, SI 1.02 at baseline and SI 5.1 after 2nd vaccination). In the healthy control group, PBMCs of 7 out of 8 subjects revealed relevant proliferation after vaccination (Appendix A).

Accordingly, baseline TBE-specific proliferation was significantly higher in the patient group compared to the healthy control group (median SI 4.2 for patients vs. 0.9 for controls, Wilcoxon Test *p* = 0.002). However, upon incubation with CEFT and PMA/ionomycin the opposite effect was observed. PBMCs of healthy controls showed a significantly stronger proliferation upon polyclonal antigen-specific stimulation with CEFT (median SI 53.9 for controls vs. 8.6 for patients, *p* < 0.001) and upon polyclonal stimulation with PMA plus ionomycin (median SI 263.2 for controls vs. 72.2 for patients, *p* = 0.01) than HSCT patients at baseline (Table 1). Notably, we found a significant and positive correlation between TBE and CEFT proliferation at baseline for the patient group (Pearson correlation coefficient: 0.71, *p* = 0.001) (Appendix A).

Very similar to the TBE-specific proliferation of PBMCs, patients’ PBMCs elaborated significantly higher amounts of cytokines upon TBE-specific activation when compared to PBMCs of healthy control subjects (Figure 2). Of note, patients’ PBMCs secreted significantly higher amounts of IL-2 (median 4.9 (range 0.6; 146.6) for patients vs. median 1 (range 0.6; 2.4) for controls, *p* = 0.03), IL-13 (median 1 (range 0.96; 98.9) for patients vs. median 0.6 (range 0.6; 0.8) for controls, *p* < 0.001), TNF-α (median 1.4 (range 0.3; 47.4) for patients vs. 0.7 (range 0.3; 1.9) for controls, *p* = 0.03) and GM-CSF (median 1.3 (range 0.3; 111.9) for patients vs. 0.7 (0.3; 1) for controls, *p* = 0.01). The cytokine responses after the second and third vaccinations did not significantly differ between patients and controls.

#### 3.2.2. Comparison of Patients with and without Humoral Response

Humoral response was previously defined by the authors as an NT ≥ 10 and an at least two-fold increase in NTs or an NT above the highest level of measurement four weeks after two TBE vaccinations [8]. According to this definition only 6 out of 17 patients had a humoral response. All six patients with a TBE-specific humoral response exhibited lymphocyte proliferation after TBE and CEFT stimulation at baseline. Lymphocyte proliferation data (SI) for all three time points stratified for humoral responders (R) versus humoral non-responders (NR) are displayed in Figure 3. Humoral responder PBMCs showed a significantly stronger baseline TBE proliferation than non-responders (median SI 27.9 for responders vs. 1.5 for non-responders, Wilcoxon Test *p* < 0.001). Additionally, in univariate logistic regression analyses, TBE proliferation showed a significant influence on humoral response after two vaccinations (*p* < 0.001). The same was observed after stimulation with CEFT. PBMCs of humoral responders also revealed significantly stronger baseline antigen-specific proliferation upon incubation with CEFT (median SI 24.6 for responders vs. 2.2 for non-responders, *p* = 0.005) (Appendix A), whereas there was no significant difference in lymphocyte proliferation after pharmacological stimulation with PMA plus ionomycine (Appendix A) which was included as an antigen-non-specific positive control for the proliferative capability of T-cells (median SI 72.5 for responders vs. 72.2 for non-responders, *p* = 0.7). Additionally, after the second vaccination, TBE-induced proliferation was significantly stronger in the group of responders (median SI 35.8 for responders vs. 1.7 for non-responders, *p* = 0.002) and there was a trend towards stronger proliferation after the third vaccination (median SI 23.8 for responders vs. 2.7 for non-responders, *p* = 0.2) (Appendix A).

As shown in Figure 4, patients with a humoral response after two vaccinations also had significantly higher cytokine responses upon TBE stimulation at baseline: IFN-γ (median 58.4 for responders vs. 1.4 for non-responders, *p* = 0.003), IL-2 (median 20.1 for responders vs. 1.2 for non-responders, *p* < 0.001), IL-10 (median 8.8 for responders vs. 1 for non-responders, *p* = 0.04), IL-13 (median 23.6 for responders vs. 1 for non-responders, *p* < 0.001), TNF-α (median 7.2 for responders vs. 1.1 for non-responders, *p* = 0.002), and GM-CSF (median 18.8 for responders vs. 1 for non-responders, *p* = 0.001). The IL-13 levels at baseline separated future TBE vaccine responders from non-responders most accurately, since all responders produced relevant amounts of IL-13, while all non-responders failed to do so (IL-13: 6/6 responders vs. 0/11 non-responders, Fishers exact test *p* < 0.001). Other cytokines had a comparable albeit lower discriminative power in separating responders from non-responders (IFN-γ: 6/6 vs. 4/11, *p* = 0.04; GM-CSF: 6/6 vs. 1/11, *p* < 0.001; TNF-α: 5/6 vs. 1/11, *p* = 0.005; IL-2: 6/6 vs. 3/11, *p* = 0.01), while IL-10 failed to do so (IL-10: 4/6 vs. 3/11, *p* = 0.2).

#### 3.2.3. Comparing Patients with Related and Unrelated Donors

Next, TBE-specific PBMC responses of patients with vaccinated sibling donors were compared to those with unrelated donors of unknown vaccination status. PBMCs of patients with related donors showed a significantly higher TBE-specific proliferation at all three time points (Figure 5). At baseline, the median stimulation index was 13.1 for PBMCs of patients with related donors and 1.5 for PBMCs of patients with unrelated donors (Wilcoxon-test *p* = 0.004). PBMCs of all patients with related donors (9/9) but only PBMCs of one patient with an unrelated donor (1/8) showed relevant TBE proliferation (SI > 3) at baseline (Fishers exact test *p* < 0.001). Proliferation after CEFT stimulation was stronger in patients with related donors but the difference was not statistically significant (median SI 14.6 for PBMCs of patients with related donors vs. 2.8 for PBMCs of patients with unrelated donors) (Appendix A).

Following vaccination, TBE-induced proliferation remained significantly stronger in patients with sibling donors compared to patients with unrelated donors. After two vaccinations, we observed a median SI of 22.8 (range 1.6; 76.6) for PBMCs of patients with related donors versus a median SI 1.6 (range 0.7; 52.4) for PBMCs of patients with unrelated donors (*p* = 0.02) and after three vaccinations we detected a median SI of 25.1 (range 15.2; 109.3) for PBMCs of patients with related donors versus 2.1 (range 0.8; 14.8) for PBMCs of patients with unrelated donors (*p* = 0.002).

Although PBMCs of patients with related donors produced higher amounts of cytokines at baseline than PBMCs of patients with unrelated donors, this was not statistically significant except for IL-13 (median 13.6 for PBMCs of patients with related donors vs. 1 for PBMCs of patients with unrelated donors, *p* = 0.04). After two vaccinations the amount of most cytokines was significantly higher in the group of patients with related donors. IL-13 was the only cytokine with significantly stronger responses between patients with related compared to unrelated donors at all three time points (*p* < 0.05) (Figure 6).

### 3.3. Predictors for TBE Proliferation at Baseline before Vaccination

Considering that PBMCs of 10 of 17 patients (58.8%) showed significant TBE-specific proliferation (SI > 3) before vaccination, which is a significant predictor for later humoral response to vaccination, we aimed to assess predictors for TBE proliferation at baseline. Table 2 summarizes the results of the univariable logistic regression model with Firth correction for the dependent variable TBE proliferation at baseline SI > 3. We assessed the variables humoral responder, related donor, sex, chronic GvHD requiring immunosuppressive therapy, BMI, age and total numbers of CD4+ T-cells at baseline. Using univariate logistic regression models, PBMCs of patients with humoral response as well as patients with related donors showed a significantly higher probability of TBE-specific proliferation SI > 3 at baseline. A significantly lower probability for TBE proliferation SI > 3 was found for increasing patient age. In the multivariate logistic regression analysis, only the variable “patients with related donors” remained statistically significant (adjusted odds ratio 27.8, CI 2.0—31,557.0, *p* = 0.016).

## 4. Discussion

This prospective single-center study is the first to investigate the cellular immune response, in particular the lymphocyte proliferative and cytokine response, after TBE vaccination in patients after allogeneic HSCT and was conceptualized as a follow-up to the previous study by Harrison et al. that detected a reduced humoral response in this group of patients compared to age-matched healthy volunteers four weeks after the second vaccination (antibody response by NT at primary endpoint: 35% in patients versus 93% in healthy controls) [8]. Similarly, Einarsdottir et al. [16] detected lower seropositivity (values ≥ 12 U/mL by Enzygnost^®^ anti-TBE Virus ELISA) rates after TBE immunization in patients after allogeneic HSCT compared to healthy control subjects (77% seropositivity after 4 doses of FSME Immun^®^ in HSCT patients versus 100% of seropositivity after 3 doses of FSME Immun^®^ in healthy volunteers). The only significant predictor for humoral vaccine response in HSCT recipients in our previous study was the total amount of CD4+ T-cells in patients [8]. Considering the unique situation after allogeneic HSCT, where a donor immune system replaces the patient’s immune system, the question arose: does the humoral response depend on donor-derived immunity to TBEV?

The main findings of this study show that first, a relevant proportion of HSCT patients (59%), of whom the majority had been vaccinated before HSCT, showed TBE-specific lymphocyte proliferation and a significant Th1 and Th2 cytokine response prior re-vaccination one year after HSCT, whereas none of the unvaccinated healthy control subjects showed such a response. Second, all HSCT patients with TBE-specific antibody responses after two vaccinations (NT ≥ 10 and at least a 2-fold increase of NT) already showed high TBE-specific lymphocyte proliferation at baseline (SI > 10) suggesting that the humoral immune response after TBE-vaccination was dependent on pre-existing TBE-specific T memory cells. Third, HSCT patients with vaccinated sibling donors had a significantly stronger lymphocyte proliferative and IL-13 cytokine response than patients who were transplanted with stem cells of unrelated donors with unknown TBE-vaccination status. Additionally, a sibling donor HSCT was an independent predictor for TBE-specific proliferation (SI > 3) pre-vaccination, which implies that TBE-specific memory cells were derived from vaccinated donors.

In accordance with our previous study, that detected neutralizing antibodies in 79% of patients at baseline, 59% of patients also showed significant lymphocyte proliferation before vaccination. Therefore, immunity against TBE was clearly detectable not only on a humoral but also on a cellular level in about half of patients in this study. This appears to be of importance for an effective vaccine response because residual antibodies might decline over time after transplantation, while remaining T and B memory cells may enable a booster reaction to re-vaccination. The situation in this study population certainly applies only to an endemic region with very high vaccination rates. A recent study from Sweden, a country with much lower vaccination rates, has drawn a different picture. Only 12% of patients showed antibodies measured by ELISA before vaccination and the majority of allogeneic HSCT patients developed antibodies only after four vaccinations [16]. Therefore, TBE vaccination rates in donors and recipients prior HSCT seem to play an important role in determining re-vaccination schemes after transplantation. Patients with unvaccinated donors or donors of unknown vaccination status might require a basic immunization after HSCT with four vaccinations, as stated by a Swedish study [16]. The situation seems to be different in patients with pre-existing TBE-specific lymphocytes, where less booster vaccinations can be sufficient to elicit a humoral response. In our study, only patients with proliferating lymphocytes at baseline showed a strong humoral response after re-vaccination with two vaccines of FSME Immun^®^.

Cytokine measurements showed similar results with significantly higher amounts of Th1 (IL-2, IFN-γ, TNF-α) and Th2 (IL-13) cytokines produced by humoral responders at baseline. Interestingly, in this study, the Th2 cytokine IL-13 levels at baseline distinguished future humoral responders from non-responders most accurately, indicating that TBE-specific T helper cell memory predicts the subsequent TBE-specific humoral response. That IL-13 in the presence of CD40L stimulation is an important growth and differentiation factor for IgM, IgG4 and IgE antibody producing B cells has been shown previously [17,18]. Along those lines, a previous study that investigated T-cell-mediated immunity after routine vaccinations during the first 15 months after autologous stem cell transplantation found higher antigen-stimulated production of Th2 cytokines IL-5 and IL-13 compared to Th1 cytokines IFN-γ and TNF-α [19]. Similar findings were made in another study in allogeneic HSCT patients, in which the number of IL-13 producing T-cells, stimulated by influenza peptides, increased significantly after seasonal influenza vaccination in all HSCT patients [20].

Our findings also show that immune reconstitution in HSCT-recipients with regard to thymic production and peripheral homeostatic expansion of naïve T cells is still heavily impaired [21]. Thus, post-thymic T cell memory cells have to complement the naïve T cells’ inability to provide efficient T cell help for the induction of protective, antigen-specific B cell responses to vaccine antigens. This finding is not unexpected given the gross distortion and slow recovery of the host immune system after HSCT. What is new and can be learned from the present study is that sibling-donor post-thymic T memory cells, being specific among other antigens for TBE, appear to become much more efficiently established in the hosts and provide efficient help to B cells for specific antibody production already after two vaccinations when compared to unrelated-donor T cells. While post-thymic memory T cells will wane after some time, the induced memory B cells might live much longer and thus well-protect the vulnerable host in the post-transplant period.

In this study, lymphocyte proliferation at baseline in HSCT patients was not unique for TBE but was also observed for other vaccine- and recall-antigens from common bacterial and viral pathogens. The cellular reactivity against an antigen mix consisting of tetanus toxoid, PPD and the CEF peptide mix (CMV, EBV, Flu) gave a similar picture regarding future TBE humoral responders. Humoral TBE responders proliferated significantly stronger upon incubation with the antigen-mix compared to TBE non-responders. In total, three patients had received vaccination against tetanus after HSCT, all showed CEFT proliferation at baseline, and some patients might have had contact with viral pathogens (EBV, CMV). However, the response to the pharmacological stimulus was the same in TBE responders and non-responders. As we did not investigate the different lymphocyte subgroups after antigen-specific stimulation, we cannot completely rule out differences in the proliferative capability of T cells after stimulation by vaccine- or recall-antigens versus pharmacological stimuli. However, we would like to point out that in case of T lymphopenia, one would expect reduced proliferation with all stimuli, not just recall-antigens. Moreover, the numbers of PBMCs used within the proliferation assays were equal across the different patients, while the composition between T and B lymphocytes and monocytes may vary to a certain degree.

In this study the only significant predictor for baseline lymphocyte proliferation (SI > 3) was a sibling donor HSCT according to the multivariable logistic regression model. Other factors like chronic GvHD requiring systemic immunosuppressive treatment did not influence the lymphocyte proliferation after stimulation with TBE antigen. Interestingly, all patients with sibling donors (9/9), but only one patient with an unrelated donor (1/8) showed significant TBE-specific lymphocyte proliferation at baseline. As mentioned before, all sibling donors were vaccinated before donation of stem cells while the vaccination status of unrelated donors was unknown. This allows to suggest that pre-existing TBE-specific memory cells may have been transferred from the post-thymic repertoire of vaccinated sibling donors to HSCT recipients [22]. Indeed, previous studies have provided evidence that immunity against vaccine-preventable diseases can be transferred from donors to recipients [23]. Several studies have shown that donor immunization enhances antibody response to certain vaccine antigens in patients after HSCT [24]. This phenomenon seems to apply for vaccines based on protein (for example, tetanus) and polysaccharide-protein (for example, Hib, PCV13) antigens, while it seems not to apply for polysaccharide-based vaccines (for example, PPV23) [25,26,27]. So far, there is limited data concerning inactivated whole-virus vaccines like FSME Immun^®^. In accordance with the above-mentioned studies, our findings suggest that the vaccination status of the donor plays a crucial role in transferring immunity against tick-borne encephalitis and improving the antibody response in HSCT recipients after re-vaccination.

While donor-derived memory T cells could be responsible for the improved vaccine response after HSCT, we cannot rule out that the observed phenomenon is the reflection of reduced graft-versus-host lymphocyte responses due to sibling-donor lymphocyte transplantation, and thus may be based on remnant host memory T cells. In contrast, graft-versus-host lymphocyte responses are known to be strong and fully depleting host lymphocytes in unrelated donor HSCT, even when the graft is HLA identical [28]. When considering that all patients had full donor-chimerism (>95% CD3+ and CD33+ lymphocytes donor-derived), it seems very unlikely that a relevant number of host T cells was able to survive. Thus, while we cannot entirely exclude the possibility that plasma cells and humoral immunity persisted in the host, because plasma cells are sessile cell types which do not tend to circulate, we can at large exclude the survival of host T lymphocytes.

This study has its limitations considering the small number of participants included in a single academic center and the limited study duration per patient of 10–12 months without further follow-up. Additionally, the study only applies to an endemic setting with high TBE vaccination rates. However, it underlines the importance of considering not only the humoral but also the cellular response in patients with immunodeficiencies and supports the positive impact of donor vaccination for improved vaccine responses in HSCT patients.

## 5. Conclusions

In conclusion, in a TBE-endemic setting with high vaccination coverage of the population, a relevant proportion of patients showed TBE-specific lymphocyte proliferative and cytokine responses before vaccination after allogeneic HSCT, which predicted a relevant humoral response in patients already after the first two vaccinations. Thus, in an endemic setting with a high rate of vaccinated donors, a conventional basic immunization consisting of three immunizations might be sufficient for patients after related allogeneic HSCT. Additionally, patients with vaccinated sibling donors were more likely to elicit a cellular immune response than patients with unrelated donors of unknown vaccination status. Therefore, our data suggest that donor immunity against TBEV represents an advantage for the HSCT recipient and a booster vaccination could be offered to related donors prior to stem cell donation to improve immunity against TBEV in HSCT recipients posttransplant.

## Figures and Tables

**Figure 1 vaccines-09-00908-f001:**
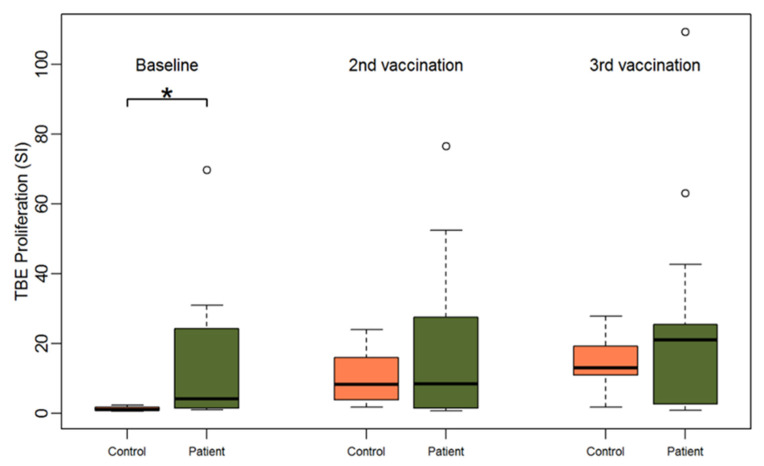
Boxplots showing the median and distribution of lymphocyte proliferation as detected by thymidine incorporation assay (the stimulation indices are given) at three different time points for healthy controls (orange) and patients (green) after stimulation with TBE antigen. Asterisk marks significant *p*-value (*p* = 0.0019).

**Figure 2 vaccines-09-00908-f002:**
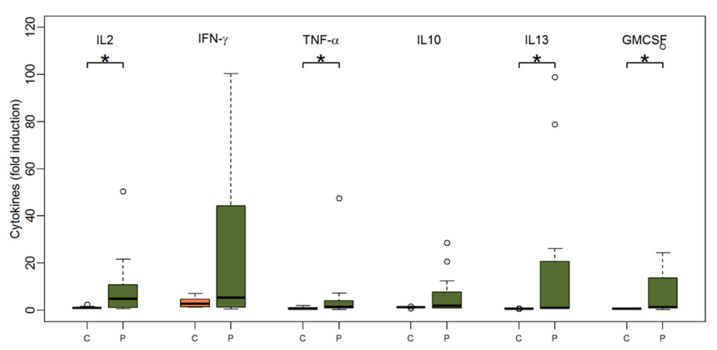
Boxplots showing the median and distribution of cytokines as detected by Luminex assay at baseline before vaccination for healthy controls (orange) and patients (green) after stimulation with TBE antigen. Asterisk marks significant *p*-value (IL2: *p* = 0.03; TNF-alpha: *p* = 0.03; IL13: *p* < 0.001; GM-CSF: *p* = 0.01).

**Figure 3 vaccines-09-00908-f003:**
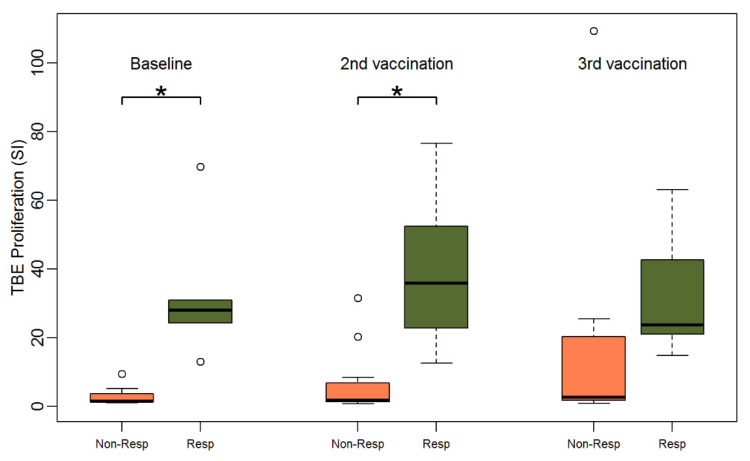
Boxplots showing the median and distribution of lymphocyte proliferation as detected by thymidine incorporation assay (the stimulation indices are given) at three different time points for humoral non-responders (orange) and responders (green) after stimulation with TBE antigen (response was defined as NT ≥ 10 and at least two-fold increase in titer after two vaccinations). Asterisk marks significant *p*-value (baseline: *p* < 0.001; 2nd vaccination: *p* = 0.0019).

**Figure 4 vaccines-09-00908-f004:**
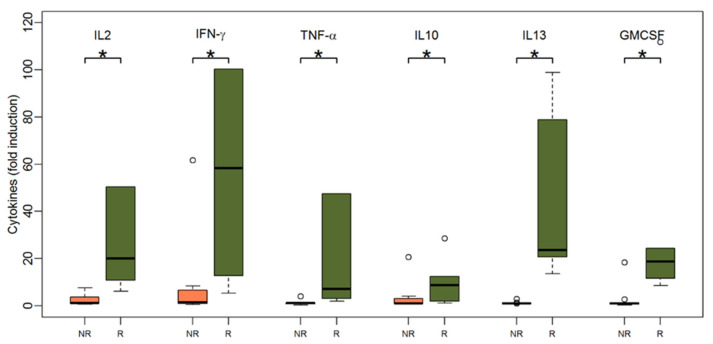
Boxplots showing the median and distribution of cytokines as detected by Luminex assay at baseline before vaccination for humoral non-responders (orange) and responders (green) after stimulation with TBE antigen (response was defined as NT ≥ 10 and at least two-fold increase in titer after two vaccinations). Asterisk marks significant *p*-value (IL2: *p* < 0.001; IFN-gamma: *p* = 0.003; TNF-alpha: *p* = 0.002; IL10: *p* = 0.04; IL13: *p* < 0.001; GM-CSF: *p* = 0.001).

**Figure 5 vaccines-09-00908-f005:**
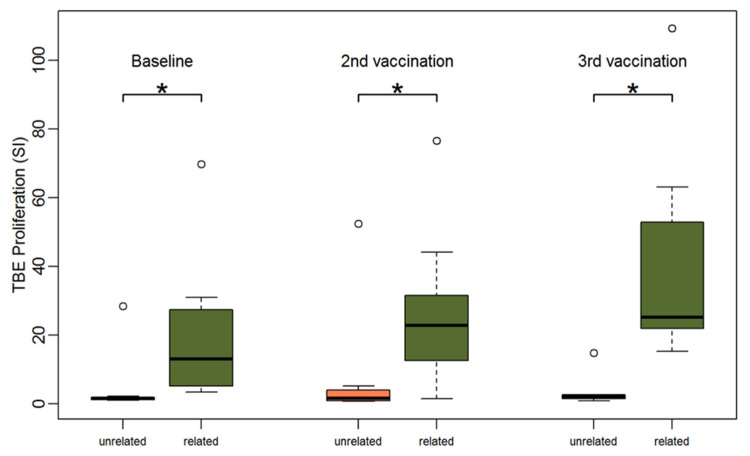
Boxplots showing the median and distribution of lymphocyte proliferation as detected by thymidine incorporation assay (the stimulation indices are given) at three different time points for patients with unrelated (orange) and related donors (green) after stimulation with TBE antigen. Asterisk marks significant *p*-value (baseline: *p* = 0.004; 2nd vaccination: *p* = 0.015; 3rd vaccination: *p* = 0.002).

**Figure 6 vaccines-09-00908-f006:**
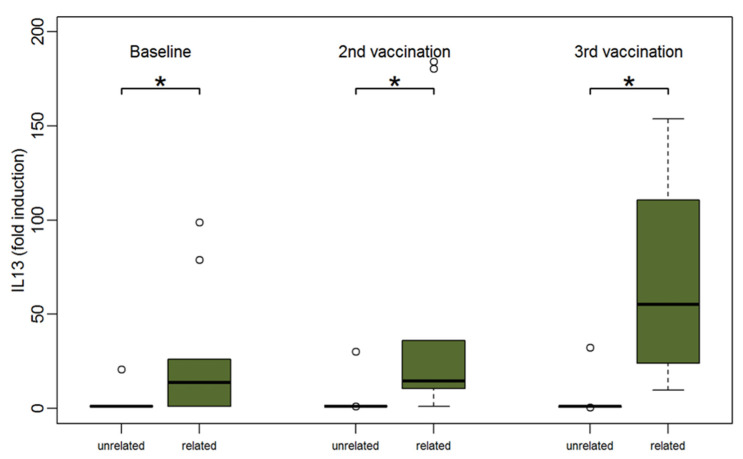
Boxplots showing the median and distribution of IL-13 as detected by Luminex assay at three different time points for patients with unrelated (orange) and related donors (green) after stimulation with TBE antigen. Asterisk marks significant *p*-value (baseline: *p* = 0.04; after 2nd vaccination: *p* = 0.01; after 3rd vaccination: *p* = 0.01).

**Table 1 vaccines-09-00908-t001:** Demographics and baseline (before vaccination, one year after HSCT) lymphocyte proliferation detected by thymidine incorporation assay (the stimulation index is given) of individual subjects.

Subject No.	Group	Age In Years	Sex	BMI	Underlying Condition	Sibling Donor	Conditioning Regimen *	Neutralization Test Titer Baseline	Lymphocyte Proliferation after Stimulation With (SI)			Humoral Responder
									TBE Antigen	CEFT	PMA/Iono	
1	Patient	27	F	23.1	AML	Yes	Myeloablative	10	30.99	24.60	67.64	Yes
2	Patient	24	M	43.1	AML	Yes	Myeloablative	160	5.15	1.53	40.28	No
3	Patient	27	M	24.0	ALL	Yes	Myeloablative	95	4.15	2.61	60.44	No
4	Patient	56	M	31.5	AML	No	Non-myeloablative	40	2.22	1.13	66.28	No
5	Patient	61	M	29.0	AML	Yes	Non-myeloablative	48	13.06	24.65	77.40	Yes
6	Patient	27	F	39.1	AML	No	Non-myeloablative	113	28.45	21.69	54.37	Yes
7	Patient	47	M	29.9	Lymphoma	No	Non-myeloablative	6	1.02	36.13	322.35	No
8	Patient	22	F	19.5	AML	Yes	Myeloablative	538	24.34	34.67	132.59	Yes
9	Patient	53	F	22.7	AML	No	Non-myeloablative	12	1.53	1.24	72.24	No
10	Patient	43	M	27.8	AML	No	Myeloablative	40	1.21	2.21	71.54	No
11	Patient	22	M	23.1	AML	No	Myeloablative	40	1.04	3.47	94.69	No
12	Patient	26	M	26.9	AA	Yes	Non-myeloablative	10	3.35	14.59	153.72	No
13	Patient	41	F	24.2	Lymphoma	Yes	Non-myeloablative	10	9.42	1.57	532.10	No
14	Patient	55	M	28.4	AML	No	Non-myeloablative	4	1.45	1.39	64.81	No
15	Patient	58	M	32.4	AML	No	Non-myeloablative	80	1.44	8.55	190.69	No
16	Patient	26	F	19.1	AML	Yes	Myeloablative	135	69.70	45.20	171.94	Yes
17	Patient	31	M	23.4	ALL	Yes	Myeloablative	4	27.41	11.21	61.41	Yes
18	Control	27	F	18.5	-	-	-	<5	0.98	134.64	271.29	Yes
19	Control	32	M	24.2	-	-	-	<5	0.87	51.46	214.57	Yes
20	Control	22	F	20.3	-	-	-	<5	0.89	77.62	374.98	Yes
21	Control	21	F	26.0	-	-	-	<5	0.99	65.14	307.39	Yes
22	Control	23	M	22.2	-	-	-	<5	2.31	13.34	314.67	Yes
23	Control	38	F	20.9	-	-	-	<5	0.61	25.19	88.62	Yes
24	Control	33	M	21.1	-	-	-	<5	1.27	45.24	85.41	Yes
25	Control	60	M	26.6	-	-	-	<5	2.37	56.29	255.10	No

Lymphocyte proliferation has been standardized based on unstimulated control samples and is shown as stimulation index (SI). * Conditioning regimen: myeloablative (*n* = 8): cyclophosphamide and full body irradiation (fractional 13.2 Gy) (*n* = 7) or cyclophosphamide plus busulfan without body irradiation (*n* = 1); non-myeloablative (*n* = 9): fludarabine-based protocols (*n* = 8) +/− cyclophosphamide (*n* = 6), +/− ATG (*n* = 3), +/− 4 Gy full body irradiation (*n* = 5) or cyclophosphamide and ATG (*n* = 1). AML: acute myeloid leukemia; ALL: acute lymphatic leukemia; AA: aplastic anemia; TBE: tick borne encephalitis antigen; CEFT: CMV, EBV, influenza virus and *Clostridium tetani* antigen mix; PMA/Iono: phorphole myristate acetate plus ionomycin.

**Table 2 vaccines-09-00908-t002:** Univariable logistic regression model with Firth correction for the dependent variable TBE proliferation with stimulation index > 3 at baseline (patients only).

Logistic Regression Models for TBE Proliferation SI > 3					
Variable	Comparison	OR	Lower 95% CI	Upper 95% CI	*p*-Value
Humoral responder	Yes vs. No	21.667	1.816	3090.366	0.012
Donor	Related vs. Unrelated	95.000	6.437	15,251.370	<0.001
Sex	Female vs. Male	4.333	0.594	52.665	0.153
Chronic GvHD requiring systemic immunosuppressive treatment	Yes vs. No	1.667	0.247	11.718	0.595
BMI		0.981	0.847	1.134	0.784
Age		0.918	0.835	0.988	0.020
Total CD4+ T-cell count at baseline		1.004	0.997	1.014	0.241

SI: stimulation index; OR: odds ratio; CI: confidence interval; BMI: body mass index; GvHD: graft-versus-host disease.

## Data Availability

The data that support the findings of this study are available at Mendeley data (https://data.mendeley.com/datasets/ps6zdcbn9b/1). The full trial protocol is available at clinicaltrials.gov (https://clinicaltrials.gov/ct2/show/NCT01991067).

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
