# Peer review of "Tick-Borne Encephalitis Specific Lymphocyte Response after Allogeneic Hematopoietic Stem Cell Transplantation Predicts Humoral Immunity after Vaccination"

_vaccines, 2021, doi:10.3390/vaccines9080908_

Round 1
Reviewer 1 Report
This MS provides a well-presented clinical study of interest to anyone working in, or subject to, HSCT. The finding that sibling donors with pre-existing TBE vaccination status is a significant predictor of the patient's response to subsequent vaccination, is beneficial. I have a couple of questions for the authors:
lines 369-371: it would be easier for the reader if the question was posed like a question i.e. 'does the humoral response depend on ....etc'?
lines 103-104: Can the authors give more information about the TBE vaccine? presumably the 2.4ug dose mentioned is of inactivated viral protein? In the context of the discussion (lines 453-462) of different vaccine types, it would be useful to know more about the TBE vaccine -any adjuvant/excipients/ viral load
line 480 (minor) -suggest insert 'to' between prior and donation
Finally, is there an intention to follow up this patient cohort for a longer-term, to see whether the non-responders do eventually respond?
Reviewer 2 Report
Do you have absolute lymphocyte counts and phenotypic analysis of the patients at the different timepoints? Is it possible that the differences you see in antigen-specific stimulation (but not in pharmacologic stimulation with PMA/Iono) merely reflects random variation due different lymphocyte populations?
You acknowledge (table 1 of your first paper, ref 8) that the majority of patients had mild chronic GVHD, Did you study chronic GVHD as a variable that could influence the SI and cytokine production?
On your first paper (ref 8) Table 2 provided the NT titer pre-HSCT and its decline post-HSCT. Do you have the equivalent T cell immunity data?
How could you differentiate between donor-transferred immunity and persistent recipient immunity? You mention this problem in the next-to-last paragraph of the Results section, but don't seem to offer any solution.
Related to this, could you provide the specific conditioning regimens used? As you know, some non-myeloablative regimens may be extremely lymphodepleting. Was serotherapy used (ATG or alemtuzumab)?
I am not sure whether in your conclusion you are suggesting HSCT recipients should be tested by TBE lymphocyte proliferation assay before deciding on the vaccination schedule. You also suggest that vaccination of the donors may be a good idea, but this is not very practical. Isn't it easier just to give 4 doses, as suggested by Einarsdottir et al?
What actionable information did you obtain and present, and what are the actions you suggest based on your findings?
Round 2
Reviewer 2 Report
Thank you for answering my questions and addressing my comments..